# Increased Indoleamine 2,3-Dioxygenase 1 (IDO-1) Activity and Inflammatory Responses during Chikungunya Virus Infection

**DOI:** 10.3390/pathogens11040444

**Published:** 2022-04-07

**Authors:** Thiara Manuele Alves de Souza, Caroline Fernandes-Santos, Jéssica Araújo da Paixão de Oliveira, Larissa Cristina Teixeira Tomé, Victor Edgar Fiestas-Solórzano, Priscila Conrado Guerra Nunes, Gabriel Macedo Costa Guimaraes, Juan Camilo Sánchez-Arcila, Iury Amâncio Paiva, Luís Jose de Souza, Paulo Vieira Damasco, Válber da Silva Frutuoso, Manoela Heringer, Luzia Maria de Oliveira-Pinto, Roberta Olmo Pinheiro, Flavia Barreto dos Santos, Elzinandes Leal de Azeredo

**Affiliations:** 1Viral Immunology Laboratory, Oswaldo Cruz Institute, Oswaldo Cruz Foundation (Fiocruz), Rio de Janeiro 21040-360, Brazil; thiara.biomed@gmail.com (T.M.A.d.S.); carol.uned@gmail.com (C.F.-S.); larissacristinateixeiratome@gmail.com (L.C.T.T.); vicfiso@gmail.com (V.E.F.-S.); priscila.nunes87@gmail.com (P.C.G.N.); gabrielmcguimaraes@gmail.com (G.M.C.G.); sanchezarcila.jc@gmail.com (J.C.S.-A.); iury.iap@gmail.com (I.A.P.); lpinto@ioc.fiocruz.br (L.M.d.O.-P.); flaviab@ioc.fiocruz.br (F.B.d.S.); 2Leprosy Laboratory, Oswaldo Cruz Institute, Oswaldo Cruz Foundation (Fiocruz), Rio de Janeiro 21040-360, Brazil; jessicapaixao-@hotmail.com (J.A.d.P.d.O.); robertaolmo@gmail.com (R.O.P.); 3Reference Center for Immune and Infectious Diseases (CRDI), Faculty of Medicine, Campos dos Goytacazes, Brazil Campos dos Goytacazes, Rio de Janeiro 28025-496, Brazil; luizjosedes@gmail.com; 4Department of General Medicine, Medicine and Surgery School, Gaffrée Guinle University Hospital, Federal University of State of Rio de Janeiro (UniRio), Rio de Janeiro 20270-004, Brazil; paulovieiradamasco@gmail.com; 5Laboratory of Immunopharmacology, Oswaldo Cruz Institute, Oswaldo Cruz Foundation (Fiocruz), Rio de Janeiro 21040-360, Brazil; vsfrutuoso@gmail.com; 6Brain Biomedicine Laboratory, Paulo Niemeyer State Brain Institute, Rio de Janeiro 20231-092, Brazil; manoelaheringer@yahoo.com.br

**Keywords:** chikungunya, indoleamine 2,3-dioxygenase 1, cytokines, chemokines

## Abstract

Chikungunya virus (CHIKV) infection causes intense cytokine/chemokine inflammatory responses and debilitating joint pain. Indoleamine2,3–dioxygenase 1 (IDO-1) is an enzyme that initiates the tryptophan degradation that is important in initial host innate immune defense against infectious pathogens. Besides that, IDO-1 activation acts as a regulatory mechanism to prevent overactive host immune responses. In this study, we evaluated IDO-1 activity and cytokine/chemokine patterns in CHIKV patients. Higher IDO-1 (Kyn/Trp ratio) activation was observed during the early acute phase of CHIKV infection and declined in the chronic phase. Importantly, increased concentrations of Tumor Necrosis Factor-α (TNF-α), Interleukin-6 (IL-6), Interferon γ (IFN-γ), C-C motif chemokine ligand 2/Monocyte Chemoattractant Protein-1 (CCL2/MCP-1) and C-X-C motif chemokine ligand 10/Interferon Protein-10 (CXCL10/IP-10) were found in the acute phase of infection, while C-C motif chemokine ligand 4/Macrophage Inflammatory Protein 1 β (CCL4/MIP-1β) was found at increased concentrations in the chronic phase. Likewise, CHIKV patients with arthritis had significantly higher concentrations of CCL4/MIP-1β compared to patients without arthritis. Taken together, these data demonstrated increased IDO-1 activity, possibly exerting both antiviral effects and regulating exacerbated inflammatory responses. CCL4/MIP-1β may have an important role in the persistent inflammation and arthritic symptoms following chikungunya infection.

## 1. Introduction

During viral infections, the innate immune response is activated by the recognition of viral nucleic acids through pattern recognition receptors (PRRs), leading to activation of innate immunity cells and antiviral responses. Innate antiviral responses culminate with the production of type I interferons (IFN) and inflammatory mediators [1]. Inflammation is the body’s protective response to ensure removal of harmful stimuli, as well as a repair process for injured tissues. Moreover, an inflammatory response can be acute or chronic, impacting the disease pathology [2]. Indoleamine 2,3-dioxygenase 1 (IDO-1) is an enzyme implicated in the host innate antimicrobial defense and immune regulation and initiates the oxidative degradation of essential amino acids such as tryptophan along the kynurenine pathway, being an important component of innate response against infections [3]. IDO-1 is expressed by macrophages, epithelial cells, and dendritic cells (DCs) and is upregulated in response to inflammatory response, especially in response to Interferon- γ (IFN-γ) [4]. The cells expressing IDO-1 are able to reduce the inflammation through cell proliferation inhibition. A combination of the effects of tryptophan depletion and the direct action of kynurenine has the potential to induce Regulatory T cells (Treg cells) to control reactive immune cells [5].

Chikungunya virus (CHIKV) belongs to the genus *Alphavirus* and represents a major arthritogenic arbovirus, causing epidemics in Brazil. Indeed, it has been responsible for important epidemics in several tropical and temperate regions of the world, and in Brazil, the first autochthonous cases were identified in the Northern and Northeast regions in 2014 [6]. Since then, the country has experienced co-circulation of CHIKV, dengue (DENV), and zika (ZIKV) viruses [7,8].

The signs and symptoms of CHIKV infection are similar to those observed in DENV infection, such as fever and myalgia. However, CHIKV infection is often associated with severe polyarthralgia. Furthermore, after the acute phase of the disease, joint symptoms may persist for weeks or even months, and some patients develop chronic joint symptoms similar to rheumatoid arthritis (RA) [9]. In addition, atypical and severe manifestations such as meningoencephalitis, encephalopathy, seizure, Guillan Barré syndrome (GBS), and neuropathies were reported [10,11,12].

The mechanisms associated with severe manifestations and/or progression to the chronic phase of the disease during CHIKV infection are poorly understood. After the mosquito bite, CHIKV enters the skin and moves into the bloodstream; infects and replicates in fibroblasts, macrophages, endothelial cells and epithelial cells. Although the infection induces a robust immune response, characterized by the production of type I IFN, recruitment of innate and adaptive immunity cells with production of neutralizing antibodies, chronic arthritis develops in some individuals. It has been suggested that the chronic polyarthralgia is due to inflammation and tissue damage caused by the host immune responses to infection [9]. Indeed, monocytes, macrophages, and natural killer (NK) cells are the main innate cellular components of inflammatory infiltrates in animal models, indicating the participation of these cells in the pathogenesis of arthritis induced by alphaviruses [13]. 

Cytokines and chemokines production promotes recruitment of immune cells to the site of infection, and these mediators may cause dysregulated inflammation. Moreover, these mediators are associated with persistent arthralgia, and it has been suggested that an imbalance of inflammatory responses during antiviral defenses might contribute to the pathogenesis of CHIKV infection [14]. Indeed, increased levels of interleukin (IL)-1α, granulocyte macrophage colony-stimulating factor (GM-CSF), C-C motif chemokine ligand 2 /Monocyte Chemoattractant Protein-1 (CCL2/MCP-1), and IL-6 have been correlated with disease severity [15,16]. We, and others, have identified inflammatory factors associated with immune cell activation during acute CHIKV infection [17,18]. 

The IDO-1 involvement during arboviral infections is unknown. During DENV infection, low levels of tryptophan and increased IDO-1 activity were observed in patients with mild dengue disease. In addition, in vitro DENV-infected DCs showed increased IDO-1 expression, and the enzyme was associated with antiviral activity [19]. Interestingly, Sun et al. [20] showed that IDO-1 inhibition in ZIKV-infected primary DCs resulted in downregulation of type I IFN responses. We have shown increased IDO-1 expression in monocytes from DENV-infected patients [21]. However, the IDO-1 activity in CHIKV-infected patients has not yet been examined.

From 2016 to 2018, Brazil experienced an explosion of CHIKV outbreaks, with thousands of human cases occurring annually; however, since February of 2020, the country has also faced the COVID-19 pandemic, which led health authorities to divert all their efforts to deal with this new challenge. Nevertheless, in 2020 and 2021, more than 80,000 and 36,242 probable cases of chikungunya were reported, respectively, in the country [22,23]. In a scenario of explosive epidemics and concomitant circulation of multiple viruses, it is essential to improve our understanding of various aspects of arboviral diseases, including the host immune factors that play a critical role in disease pathogenesis and clinical outcomes. Studies on host immune responses during acute and chronic chikungunya in humans are still limited. 

Considering that IDO-1 activation acts as regulatory mechanism to prevent overactive host immune responses, we evaluated if patients with a higher IDO-1 activity would have a reduced inflammatory activity, and which correlated with reduced symptoms in a disease–phase manner. Therefore, in this study we determined the IDO-1 activity, and cytokine and chemokine patterns in CHIKV-infected patients during acute, post-acute, and chronic phases of infection. The knowledge of the natural immune response triggered by CHIKV infection will certainly contribute to the development of effective vaccines and treatments. 

## 2. Results

### 2.1. Demographic, Clinical, and Laboratorial Characteristics of Chikungunya-Infected Patients

During the 2016 and 2018 CHIKV outbreaks, a total of 353 suspected cases of arbovirus infection was investigated. According to the Ministry of Health’s guidelines, patients who presented fever and arthralgia were considered as suspected cases [24]. All plasma samples were submitted to differential diagnosis by serological and molecular assays for confirmation or exclusion of CHIKV infection. 

Eighty-eight patients (88/353; 24.9%) were included in the study. Eight-three patients (83/353, 23%) presented a positive molecular test (RT-qPCR) for CHIKV and negative for DENV, ZIKV, and Mayaro (MAYV). In addition, five patients tested negative for CHIKV, DENV, ZIKV, and MAYV, but were positive for anti-CHIKV IgM ELISA, and, therefore, were included in the study. Twelve paired samples were included in the analyses, and, thus, 100 plasma samples were evaluated.

The patients were categorized into groups according to the duration of symptoms: acute (up 14 days, *n* = 78), post-acute (14–90 days, *n* = 12), and chronic (91–124 days, *n* = 10) [25], and they were classified according to the Ministry of Health’s guidelines [24]. All plasma samples from acute patients had detectable CHIKV viral RNA. Five plasma samples (41%) from post-acute patients and most patients at chronic phase (80%) had detectable CHIKV viral RNA.

The median age of acute CHIKV patients was 45 years-old (range 28–57) and 45 (57%) of them were female. Forty-two (53%) had a pre-existing condition; the most frequent comorbidities were hypertension (26%). Less frequent comorbidities were rhinitis (6%), chronic obstructive pulmonary disease (1%), and cardiac disease (1%). 

Most acute patients enrolled presented fever (92%), frequently accompanied by joint pain. Signs/symptoms such as arthralgia (93%), myalgia (69%), headache (73%), prostration (75%), edema (51%), exanthema (52%), low back pain (61%), anorexia (67%), and pruritus (43%) were also reported. Vomiting (21%) and abdominal pain (16%) were less frequently observed among acute CHIKV cases.

The median age of patients with post-acute chikungunya was 54 years old (range 36.7–65) and 58% were female. The post-acute patients enrolled presented signs/symptoms such as arthralgia (100%), myalgia (50%), headache (66%), prostration (50%), and edema (58%). Fever (25%) was less frequently reported among post-acute CHIKV cases, and the most frequent comorbidities observed were hypertension (20%) and diabetes (10%).

Twenty-nine (29/88; 33%) patients had polyarthritis in the acute and post-acute phases of infection. The median age of was 50 years-old (range 34–59), and 58% of them were female. The polyarthritis involved small and large joints (hands, knees, feet, ankle, and fists) and most cases had symmetrical arthritis. Three chikungunya patients (3/88; 3.4%) were hospitalized, but they were discharged alive without complications. No fatal cases were reported among the patients investigated. One patient had severe CHIKV infection with atypical neurological disorders such as seizure and confusion.

Ten (10/88; 11%) patients with clinically acute and post-acute CHIKV infection progressed to chronic polyarthralgia (i.e., pain with relapsing arthralgia for at least 3 months after initial symptoms). Of these, six (60%) patients had chronic inflammatory polyarthritis and edema. In addition, 50% of the chronic patients presented prostration, and all had arthralgia. The median age of chronic chikungunya patients was 60 years-old (range 52–66.2), and 60% of them were female. The demographic characteristics and clinical data of the chikungunya patients are summarized in Table 1.

CHIKV infection confirmation was based on the detection of specific viral RNA [26] and anti CHIKV specific IgM and IgG in the CHIKV RNA-PCR positive cases. Overall, CHIKV cycle threshold (Ct) values ranged from 13.6 to 37.2 and the median of the CHIKV viral load values was 4.5 (2.4–7.1; min/max). The CHIKV viral RNA was detected by real-time RT-qPCR in the first days after onset of illness, decreasing by 8–15 days. Remarkably, CHIKV RNA was detected by RT-qPCR in patients at the chronic phase of infection (91–124 days). Anti-CHIKV IgM antibodies were detected in the 4–7 days after onset of illness, while anti-CHIKV IgG antibodies appeared after 8–15 days. Anti-IgM and IgG could be detected in the plasma up to 3 months after onset of the disease, while the viral load decreased within days (Figure 1). Moreover, the median of CHIKV viral load during acute and post-acute phases was 4.9 copies/mL (range 3.1–6.2) and 3.0 copies/mL (range 2.9–4.1), respectively, and at chronic phase of infection was 4.1 copies/mL (range 3.7–4.2). We found significant increases in the CHIKV viral load at 1–3 days compared to 4–7 and 8–15 days after onset of illness [(1–3 days 6.0 (range 4.9–6.5) vs. 4–7 (3.6 (range 3.1–4.4), *p* = 0.001)]; [(1–3 days 6.0 (range 4.9–6.5) vs. 8–15 (3.1 (range 2.6–3.3), *p* = 0.001)].

### 2.2. Circulating Plasma Levels of Kynurenune, Tryptophan and Indoleamine 2,3 Dioxygenase (IDO-1) Activity in CHIKV-Infected Patients

IDO-1 is an enzyme involved in antiviral defense and immune regulation, and it initiates the oxidative degradation of essential amino acids, such as tryptophan along the kynurenine pathway. This aminoacid degradation is an important component of innate response against viral infections. More importantly, IDO-1 provides protection from oxidative stress caused by excessive inflammatory responses [3]. In order to investigate IDO-1 activity during CHIKV infection, circulating levels of kynurenine and tryptophan, and the IDO-1 activity (Kyn/Trp ratio) were determined using HPLC [27]. 

As observed in the Figure 2, kynurenine levels are increased during the acute phase of CHIKV infection compared to healthy donors (HD) and patients at the chronic phase of the disease (Figure 2a). In addition, acute patients presented decreased levels of Tryptophan compared to chronic patients (Figure 2c). Accordingly, acute CHIKV-infected patients also presented increased IDO-1 activity (Kyn/Trp ratio) compared to HD and chronic patients (Figure 2e). Next, we analyzed the possible associations of IDO-1 activity with acute phase, following persistent joint pain. For this purpose, we analyzed paired samples (acute phase vs. chronic phase). The data demonstrated higher Kynurenine levels (Figure 2b) and higher IDO-1 activity during the acute phase of infection (Figure 2f) compared to the chronic phase. 

We conducted a kinetic analysis from day 3 to 124 post-infection (DPI). As demonstrated in Figure A1, kynurenine plasma levels were increasing during the first days of illness, while tryptophan levels were decreasing, reaching increased levels from day 16. Consequently, the IDO-1 activity (Kyn/Trp ratio) was increased at 1–3 DPI, declining from 16 DPI. Taken together, our data demonstrate that systemic IDO-1 activity is increased in the plasma of acutely-infected CHIKV patients.

### 2.3. Measurement of Plasma Levels of Cytokines and Chemokines

A substantial activation of cytokines and chemokines is involved in the inflammatory responses in CHIKV-infected patients. Excessive inflammatory response in CHIKV infections is postulated to be a major driver of chikungunya pathogenesis in individuals with chronic infection [28]. To determine a cytokine/chemokine signature during infection, we next evaluated circulating levels of cytokines/chemokines in CHIKV-infected patients at acute, post-acute, and chronic phases of infection.

Tumor Necrosis Factor- α (TNF-α) and IL-6 concentrations were significantly higher during the acute phase of infection compared to HD (Figure 3a,c). In addition, IFN-γ levels were significantly higher during the acute and post-acute phases of infection (Figure 3e). IL-10 plasma concentrations were not statistically different, and most patients had IL-10 levels below the detection limit of the assay. As observed in Figure 3g, few acute samples showed detectable levels of IL-10. Some patients showed increased plasma levels of cytokines (TNF-α, IL-6, and IFN-γ) during the chronic phase of infection, however no statistically significant differences were found when acute and chronic cases were compared (Figure 3b,d,f,h). Overall, TNF-α and IL-6 production was increased at the acute phase of CHIKV infection (1–3 and 4–7-DPI), declining after 91 DPI (Figure A1).

We also measured the amounts of chemokines CCL2/MCP-1, C-C motif chemokine ligand 4/ Macrophage Inflammatory Protein-β (CCL4/MIP1-β),C-C motif ligand 5/ Regulated on Activation, Normal T cell Expressed and Secreted ( CCL5/RANTES), C-X-C motif ligand 8/ Interleukin- 8 (CXCL8/IL-8), and C-X-C motif chemokine ligand 10/ Interferon Protein-10 (CXCL10/IP-10) in the patients’ plasma. The concentrations of CCL2/MCP-1 and CXCL10/IP-10 were significantly higher in the CHIKV-infected patients during the acute phase of the disease in comparison to HD and patients in the chronic phase (Figure 4a,c). Generally, CCL2/MCP-1 plasma levels were increased in the acute phase of CHIKV infection, but decreased in the chronic phase, as seen in the paired-sample analyses (Figure 4b). The CXCL8/IL-8 concentrations were increased in some patients at the acute phase and the concentrations of CXCL8/IL-8 were found to be significantly higher in patients with chronic infection, as compared to HD (Figure 4e). However, no differences were observed when acute and chronic phases of infection were compared (Figure 4f). 

We found that the CCL4/MIP1-β plasma levels were significantly higher during the acute phase of infection, increasing in the chronic phase (Figure 4g). Indeed, patients in the chronic phase showed higher CCL4/MIP1-β concentrations (Figure 4h). On the other hand, CCL5/RANTES plasma levels were decreased during the acute phase of infection when compared to HD, and increased levels were found in the chronic phase of infection (Figure 4i). 

CCL2/MCP-1 and CXCL10/IP-10 showed a similar kinetic profile, with the plasma levels decreasing from day 8 to 15, with day 16–124 as the lowest level. On the other hand, CCL4/MIP1-β and CCL5/RANTES did not show such a trend, and the plasma levels were increased from 91 to 124 DPI (Figure A1).

### 2.4. Immunological Profile in CHIKV-Infected Patients with Arthritis and Without Arthritis

As mentioned previously, twenty-eight (29/33%) patients had polyarthritis in the acute and post-acute phases of infection; therefore, we analyzed the immunological profile of patients with and without arthritis. Although, patients showed significantly increased levels of TNF-α and IL-6, no statistically significant differences were found for TNF-α and IL-6 levels between patients with or without arthritis. However, IFN-γ concentrations and IDO-1 activity were significantly increased in arthritic and non-arthritic patients compared to HD (Figure 5a,b). Likewise, elevated plasma concentrations of CCL2/MCP-1 and CXCL10/IP-10 were found in arthritic and non-arthritic patients compared to HD, although the arthritic group had higher median values of CXCL10/IP-10 [HD 15 (8–19), without arthritis 38 (28–89), with arthritis 66 (44–116) median and interquartile range (IQR)] (Figure 6a,b). Remarkably, CCL4/MIP-1β plasma concentrations were significantly higher among patients with arthritis than in non-arthritic patients [(HD 242 (203–301), without arthritis 244 (81–495), with arthritis 435 (273–994) median and interquartile range (IQR)] (Figure 6c). Lastly, lower concentrations of CCL5/RANTES were seen in arthritic and non-arthritic patients compared to HD (Figure 6d).

### 2.5. Associations of Immunological and Virologic Factors

The present study also addressed the relationship of the immunological and virologic factors during the acute and post-acute phases of infection, as demonstrated in Figure 7. We found that IL-6, CCL2/MCP-1, and CXCL10/IP-10 plasma levels were directly associated with viral loads [(IL-6 (r = 0.454, *p* < 0.001); CCL2/MCP-1 (r = 0.468, *p* < 0.001) and CXCL10/IP-10 (r = 0.393, *p* < 0.013)]. In addition, IL-6 levels were directly associated with CCL2/MCP-1 and CXCL10/IP-10 as well [(CCL2MCP-1 (r = 0.558, *p* < 0.001) and CXCL10/IP-10 (r = 0.577, *p* < 0.001)]; whereas, the CCL2/MCP-1 levels were directly associated with CXCL10/IP-10 (r = 0.801, *p* < 0.001). Finally, the kynurenine levels were directly associated with CXCL-8/IL-8 concentrations (r = 0.482, *p* < 0.001) (Figure 7).

## 3. Discussion

Arboviruses, including DENV, ZIKV, and CHIKV, have impacted public health significantly. Explosive outbreaks of CHIKV, an alphavirus, in 2016, 2017, and 2018 posed new challenges for clinical differential diagnosis, as patients with dengue, zika, and chikungunya share similar clinical signs and symptoms. The spread of CHIKV in Brazil has posed a challenge for diagnosis and for disease management by rheumatologists, becoming a serious public health problem. In this study, we comprehensively analyzed immunological and virologic markers in patients with confirmed CHIKV infection, as we currently have a limited understanding of the immunologic and virologic mechanisms involved during infection, and the immunopathogenesis of CHIKV-related arthritis remains poorly understood.

This study was carried out in patients with a confirmed diagnosis of CHIKV infection and classified according to the Brazilian guidelines for management of CHIKV infection [24]. We observed a predominance of patients with mild clinical conditions and some cases evolved to a chronic phase of infection. In addition, most patients reported fever and arthralgia, respectively, as demonstrated previously [17,29]. 

IgM responses appeared within 4 days of onset of symptoms, while IgG responses were observed at 8–15 days after disease onset. Furthermore, IgM and IgG antibodies were observed in patients for 3 months, coinciding with the decrease of the CHIKV viral loads. Most of the chronic patients still had detectable CHIKV viral RNA at 91–124 DPI. Indeed, CHIKV RNA was found in perivascular synovial macrophages and fibroblasts during chronic phase [30,31,32]. The persistence of symptoms seems to be a direct consequence of the viral infection, in addition to the association with the immune response [14]. Therefore, how the virus persists, and what role it plays in the patient’s immune system and the clinical course of chronic arthralgia, are still controversial and not fully understood; thus, further studies on human cases are needed.

In the present study, we reported higher IDO activity in CHIKV-infected patients. Lower concentrations of tryptophan, besides enhanced concentrations of kynurenine and, consequently, increased IDO-1 activation were found in the plasma from acute patients compared to HD and chronic patients. Nutrient deprivation represents an important innate immune mechanism of host defense. IDO-1 is known to inhibit the proliferation of some pathogens, such as bacteria, parasites, and virus [33]. Bodaghi et al. [34] demonstrated that inhibition of viral replication of human cytomegalovirus (CMV) in primary retinal epithelial cell cultures was induced by IFN-γ and IFN-β and inhibition of CMV replication was reversed by exogenous addition of tryptophan, indicating IDO-1 involvement during infection. Other human viral infections, such as herpes simplex virus type 2 (HSV-2) and measles virus (MV) were sensitive to IDO-1-induced tryptophan depletion [35]. The antiviral activity induced by IFN-γ in the in vitro cell culture with MV was associated with the induction of IDO-1 in these cells [36]. Likewise, inhibition of HSV-2 by IFN-γ in cultured Hela cells was also mediated by IDO-1 [35]. Low circulating levels of tryptophan, and increased IDO-1 activity in patients with dengue were previously demonstrated [19,37]. Importantly, treatment of DENV-infected DCs with IDO-1 inhibitor was able to reverse the antiviral effect, with an increase in infected DCs, suggesting that IDO-1 may exert antiviral effects [19]. These results suggest an antiviral role of the enzyme during viral infections; however, some viruses are able to hijack the effects of IDO-1, using them to facilitate their own life cycle; as is the case with Human Immunodeficiency Virus (HIV) [38]. Overall, our results suggest that, in early stages of CHIKV infection, the virus is capable of inducing IDO-1 enzymatic activity that could have an anti-viral effect through tryptophan deprivation.

Consequences of IDO-1 activation include, besides inhibition of microbial growth, immunoregulatory effects, for instance, Treg cell expansion and inhibition of T-helper 17 (Th17) cells, which in turn attenuate inflammatory responses [5]. IL-10 is an anti-inflammatory cytokine that participates in immunoregulation and exerts protective effects by development of Treg cells [39]. Treg cells are essential in the alleviation and resolution of arthritis [40] and were associated with protection against the deleterious effects of CHIKV infection [41]. In addition, levels of IL-10 were found slightly and or highly increased during chikungunya infection [42]. We noted that IL-10 were detected in very few patients during the acute phase of infection, in agreement with our previous study [17]. It could be that the lack of IL-10 production observed in our study is due to a lack of control in the immune system, where there is an exacerbated production of pro-inflammatory cytokines and cell activation with the same profile and reduction of immune regulation as the Treg cell itself.

IDO-1 contributes to the development of pathology in chronic inflammatory diseases; and several studies have shown a key role of IDO-1 metabolites in RA pathogenesis. Importantly, one study found downregulation of tryptophan-related metabolomic profile in RA synovial fluid [43], while lower concentrations of tryptophan in both synovial fluid and serum were reported [44]. Studies using the experimental model of collagen-induced arthritis in rodents have shown a key role of the IDO-1 pathway in the improvement of arthritic symptoms. IDO-1 inhibition generated joint injuries and IL-17 production [45]. These results suggested that the IDO/kynurenine pathway could have a regulatory role, in an attempt to control excessive inflammation and tissue damage. However, here, no difference was observed in the kynurenine concentrations and or Kyn/Trp ratio in patients without and with arthritis.

We observed that chronic patients had lower and or normal kynurenine concentrations and Kyn/Trp ratio, since the values did not differ from healthy individuals, but were significantly decreased compared to acute patients. Chronic patients were treated with prednisolone and methotrexate, and these drugs may affect the kynurenine pathway. However, studies demonstrated that the treatment did not appear to influence the kynurenine pathway in patients with RA [46]. Importantly, acute and chronic chikungunya patients were shown to have higher levels of IL-17 [15], a pro inflammatory implicated in the etiology of RA and associated with bone tissue inflammation [45]. It is possible that low IDO activity in the chronic phase of CHIKV infection might alter the balance of Th17/Treg. Nevertheless, further studies are required to investigate the role of IDO-1 during CHIKV infection, especially in chronic patients. 

Studies suggested that the host’s immune response is crucial in controlling CHIKV infection and that dysregulated responses can be damaging to the host, as is seen in severe and chronic cases [14,30]. Several evidences have demonstrated that persistent arthralgia could be the result of the host inflammatory response and increased production of inflammatory mediators, including IFN-α, IFN-γ, TNF-α, IL-1β, IL-6, IL-7, CXCL8/IL-8, IL-10, IL-12, IL-15, IL-17, IL-27, GM-CSF, CCL2/MCP-1, and CXCL10/IP10 [47]. 

Besides the evidence for increased IDO-1 activation during CHIKV infection, our data support the observation that CHIKV infection leads to increased production of IFN-γ in agreement with previous studies [48]. IFN-γ has a critical role in eliminating pathogens, being a key molecule in the innate immunity against viral infections, and it is essential for promoting immune cell activation [49], but also contributes to tissue injuries in RA [50]. In CHIKV infection, the CD4+ T cells migrate to synovium, leading higher concentrations of IFN-γ, but these cells mediated joint inflammation independently of IFN-γ, suggesting that IFN-γ is not the single factor associated with joint pathology during infection [51]. Furthermore, IDO-1 activity is upregulated in antigen-presenting cells (APCs) in response to the IFN-γ secreted by activated T cells and probably protects the host from CHIKV infection and uncontrolled immune responses.

We found higher concentrations of TNF-α and IL-6 in the acute phase of infection. In fact, TNF-α and IL-6 are both pleiotropic cytokines playing major roles in acute inflammation [2]. Several studies reported significant increases in TNF-α, IL-6 and C-reactive protein (CRP) levels in acute and chronic patients [15,52,53]. Importantly, IL-6 levels were reported in severe patients and associated with persistent arthralgia, suggesting deleterious effects during infection [16,54,55].

Similarly to other studies, we report that patients with CHIKV infection presented a strong inflammatory response, characterized by the production of chemokines [29,30]. The first days of infection were characterized by a higher production of CCL2/MCP-1 and CXCL10/IP-10, which decreased as the infection progressed. CCL2/MCP-1 is an important chemokine in the recruitment of monocytes/macrophages and is responsible for the migration of memory T cells [56]. CXCL10/ IP-10, is a pleiotropic protein that mediates the migration and regulation of monocytes, NK cells, and T cells in response to IFN-γ [57]. In our study, both CCL2/MCP-1 and CXCL10/IP-10 were positively associated with CHIKV viral loads. Increased concentrations of CCL2/MCP-1 and CXCL10/IP-10 resulted in an augmented recruitment of monocytes to infection sites and, consequently, might generate cells more susceptible to CHIKV infection. Indeed, the use of Bindarit, an inhibitor of CCL2/MCP-1 synthesis, improved joint symptoms and reduced tissue destruction in CHIKV and Ross River Virus (RRV)-infected mice in vivo, indicating that CCL2/MCP-1 may be involved in joint pathology [58]. Of note, previous studies have demonstrated increased levels of CCL2/MCP-1 and CXCL-10/IP-10 in chronic and severe patients [47,54].

Another interesting finding was that chronic patients presented higher CCL4/MIP-1β concentrations compared to acute phase individuals. Additionally, patients with arthritis had higher CCL4/MIP-1-β concentrations. CCL4/MIP-1β coordinates acute and chronic inflammatory host responses at sites of infection and is essential in the chemotaxis and transendotelial migration of monocytes and NK cells [59]. Indeed, monocytes and macrophages are the main target cells and components of inflammatory infiltrates in experimental animal models of CHIKV infection [60]. Remarkably, NK cells produce the CCL4/MIP-1β that drives inflammatory cells to the site of infection [61]. These cells are activated during acute CHIKV infection [62,63] and are associated with viral-induced pathology [64]. NK cell depletion significantly reduced acute joint inflammation, suggesting that NK cells infiltrating synovial tissues could contribute to the development of chronic inflammation [64].

Besides CCL4/MIP-1 β, patients had higher levels of CXCL8/IL-8 in the chronic phase compared to healthy donors, confirming previously published findings [29]. In addition, kynurenine levels were directly associated with CXCL8/IL-8 levels. CXCL8/IL-8 is another pro-inflammatory mediator that mediates the recruitment of innate immune cells to inflammatory site [65] and was shown to increase kynurenine production [66].

CCL5/RANTES is a CC chemokine that is a key regulator of cell proliferation and leukocyte trafficking [65]. CCL5/RANTES induces collagen degradation in human RA synovial fibroblasts, contributing to the tissue damage [67]. Decreased concentrations of CCL5/RANTES were reported as a potential biomarker for chikungunya severity [16,55]. Indeed, we found decreased concentrations of CCL5/RANTES in the acute phase and levels of CCL5/RANTES at chronic phase were significantly higher than those at acute phase but were still not as high as those in healthy individuals. Also, there was no statistically significant difference in relation to arthritis symptoms.

Finally, plasma levels of chemokines in the acute phase of infection were mainly monocyte chemoattractants or secreted by monocytes and macrophages. IDO-1 contributes to the regulation of effector immune responses triggered by IFN-γ, especially chemokines that are IFN-dependent, such as CXCL-10/IP-10. Our study suggested that the persistence of elevated levels of CCL4/MIP-1β could be a contributor to the exacerbated and persistent inflammation and arthritic symptoms seen in CHIKV infection.

Our study has some limitations. Although we tested samples from acute and chronic patients, a low number of paired samples was analyzed. We only followed up patients for 3 months after the onset of symptoms, and the information about the evolution of arthritic symptoms in the most acute patients was not evaluated. Furthermore, we were not able to determine the pattern of Treg cells. However, the data presented here shows how increased IDO-1 activity and inflammatory responses are related to the distinct phases of infection, and this, therefore, contributes to the understanding of the mechanisms underlying chikungunya pathogenesis. Further studies with larger sample sizes are needed, to confirm the association of these factors and development of chronic disease.

## 4. Materials and Methods

### 4.1. Study Subjects

The investigation was performed during CHIKV outbreaks in Rio de Janeiro (RJ), Brazil between March 2016 and March 2018. We carried out an observational study of 353 suspected cases of CHIKV infection. Blood samples were obtained from patients assisted in two hospitals: at the Reference Center for immune and infectious diseases in Campos dos Goytacazes, RJ, and at the Hospital Rio-Laranjeiras, in the metropolitan region of RJ. Patients presenting fever and rash during the acute phase of infection, followed by at least two of the following signs and symptoms: fever, headache, myalgia or arthralgia, conjunctivitis, pruritus, retro-orbital pain, and prostration were recruited as suspicious for arboviral infection. Patients were interviewed and clinically examined by a physician at the time of hospital admission. Data were correlated to the days of disease onset until the moment when the patients were interviewed. 

CHIKV suspected cases were considered any patient with sudden onset of high fever, who presented arthralgia or severe arthritis of acute onset, not explained by other conditions, and being a resident or having visited endemic or epidemic areas up to two weeks before the onset of symptoms or that had an epidemiological link with a confirmed case [24]. CHIKV diagnosis was based on the detection of the viral genome by real-time reverse transcription PCR (RT-qPCR). The inclusion criteria consisted of a patient with a positive molecular test for CHIKV (RT-qPCR) and negative for ZIKV, DENV, and MAYV. Patients tested positive by soluble nonstructural NS1 protein for DENV were excluded. 

The cohort of CHIKV patients included those with varying clinical symptoms/signs and chronicity of symptoms. Eighty-eight CHIKV positive patients (88/353; 24.9%) were included in the study, and they were categorized into three phases: acute phase (up to 14 days after disease onset), a post-acute phase (15 to 90 days), and a chronic phase (after 90 days) [25]. Blood samples from 10 healthy donors (HD) were collected and used as controls in the analyses. At moment of blood collection, the donors had no signs/symptoms of infection for at least for 3 months.

### 4.2. Ethics Statement

Written informed consent was obtained from participants prior to any procedure. The study was approved by the National Commission of Ethics in Research (Certificate number CAAE: 56913416.9.0000.5243). 

### 4.3. Laboratorial Diagnosis

Peripheral blood was taken by means of vein puncture, using BD Vacutainer™ tubes containing acid-citrate-dextrose. Approximately 20 mL of peripheral blood was obtained. The plasma was centrifuged, aliquoted, and stored at −80 °C, until analysis.

Suspected cases of CHIKV infection were confirmed by detection of viral RNA using CHIKV specific real-time RT-qPCR [26]. Viral RNA extractions were obtained from whole blood or plasma samples using a QIAmpViral Mini Kit (Qiagen Inc., Valencia, Spain), following the manufacturer’s predetermined protocol. RNA was analyzed by quantitative polymerase chain reaction, and viral titers were determined using a standard curve. The detection of anti-CHIKV IgM and IgG antibodies was performed using ELISA kits (Chikungunya IgM EI293a-9601M; Chikungunya IgG EI293a-9601G, Euroimmun, Lübeck, Germany), according to the manufacturer’s instructions.

Differential diagnosis was performed using Dengue [68], Zika [69], and Mayaro [70] real-time RT-qPCR. Specific dengue serology tests, including Platelia dengue NS1 Ag-ELISA (BioRad Laboratories, Berkeley, CA, USA) were performed according to the manufacturer’s instructions. 

### 4.4. Measurement of Plasma Cytokines and Chemokines

The concentrations of 9 parameters, including cytokines and chemokines present in the plasma samples, were estimated using ELISA kits, in accordance with the manufacturer’s instructions. The cytokines/chemokines that were assayed included IFN-γ (cat. 900-K27, Peprotech, NJ, USA); TNF-α (cat. 900-K25, Peprotech, NJ, USA); CCL2/ MCP-1 (cat. 900-K31, Peprotech, NJ, USA); CXCL10/IP-10 (cat. 900-K39, Peprotech, NJ, USA); IL-6 (cat. DY206, R&D Systems, MN, USA); IL-10 (cat. DY217B, R&D Systems, MN, USA); CCL5/RANTES (cat. 900-K33, Peprotech, NJ, USA); CCL4/ MIP1-β (cat. 900-T36, Peprotech, NJ, USA); and CXCL-8/ IL-8/ (DY208, R&D Systems, MN, USA). Standard curves of known concentrations of recombinant human cytokines or chemokines were used to convert optical density (OD) into concentration units (pg/mL). The levels of cytokines/chemokines were analyzed using a SpectraMax Paradigm^®^ instrument (Molecular Devices, San Jose, CA, USA).

### 4.5. Measurement of the IDO-1 Enzymatic Activity

IDO-1 measurement was performed through high performance liquid chromatography analysis (HPLC). The IDO-1 activity was evaluated by the ratio between the values of kynurenine and tryptophan (Kyn/Trp) present in the plasma of CHIKV-infected patients, according Salles et al. [27].

### 4.6. Statistical Analysis

Kynurenine and tryptophan concentrations, IDO-1 activity as well as cytokine/chemokine concentrations between groups were calculated using the Kruskal-Wallis test followed by Dunn’s multiple comparisons test. Spearman’s rank correlation between the variables studied was represented in a correlogram. Wilcoxon matched-pairs signed rank and Mann–Whitney U-test were used to compare paired and unpaired analyses. Statistical analyses were conducted in R Statistical Language version R-4.04 for Windows (R Core Team [2020]. R: A language and environment for statistical computing and R Foundation for Statistical Computing, Vienna, Austria. URL: https://www.R-project.org accessed on 20 April 2021); and using a GraphPad Software, version 6. *p* values of <0.05 were considered statistically significant.

## 5. Conclusions

Our findings demonstrated that the acute phase of CHIKV infection was characterized by increased IDO-1 activity and inflammatory responses in CHIKV-infected patients, suggesting an important role in the innate antiviral response during infection. To the best of our knowledge, this is the first report to describe IDO-1 responses in CHIKV infection. These responses may control the tissue damage and/or modify the inflammatory response elicited by the CHIKV infection. A better understanding of the regulation of the human immune response and the role of Tregs cells during infection may provide new tools for effective treatments. 

## Figures and Tables

**Figure 1 pathogens-11-00444-f001:**
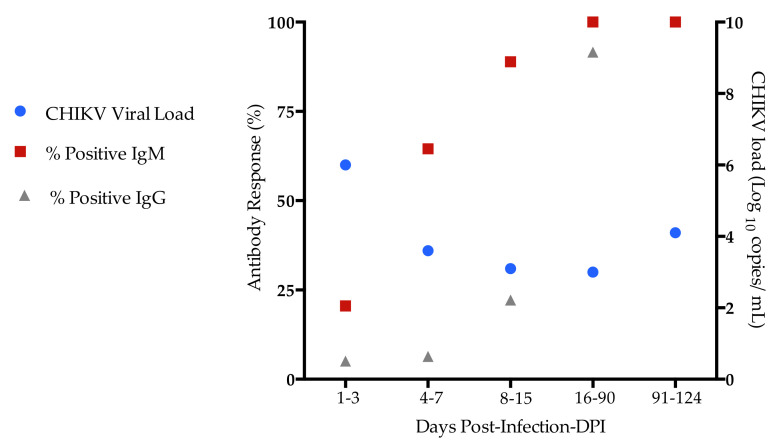
Kinetics of the CHIKV viral load and specific anti-CHIKV antibodies in chikungunya patients, according to days post-infection (DPI). The y-axis represents percentages of positive CHIKV-specific IgM and IgG in plasma of infected patients at 1–3, 4–7, 8–14, 15–90, and 91–124 DPI. The x-axis represent viremia (median of viral RNA copies) in plasma of infected patients at 1–3, 4–7, 8–14, 15–90, and 91–124 DPI.

**Figure 2 pathogens-11-00444-f002:**
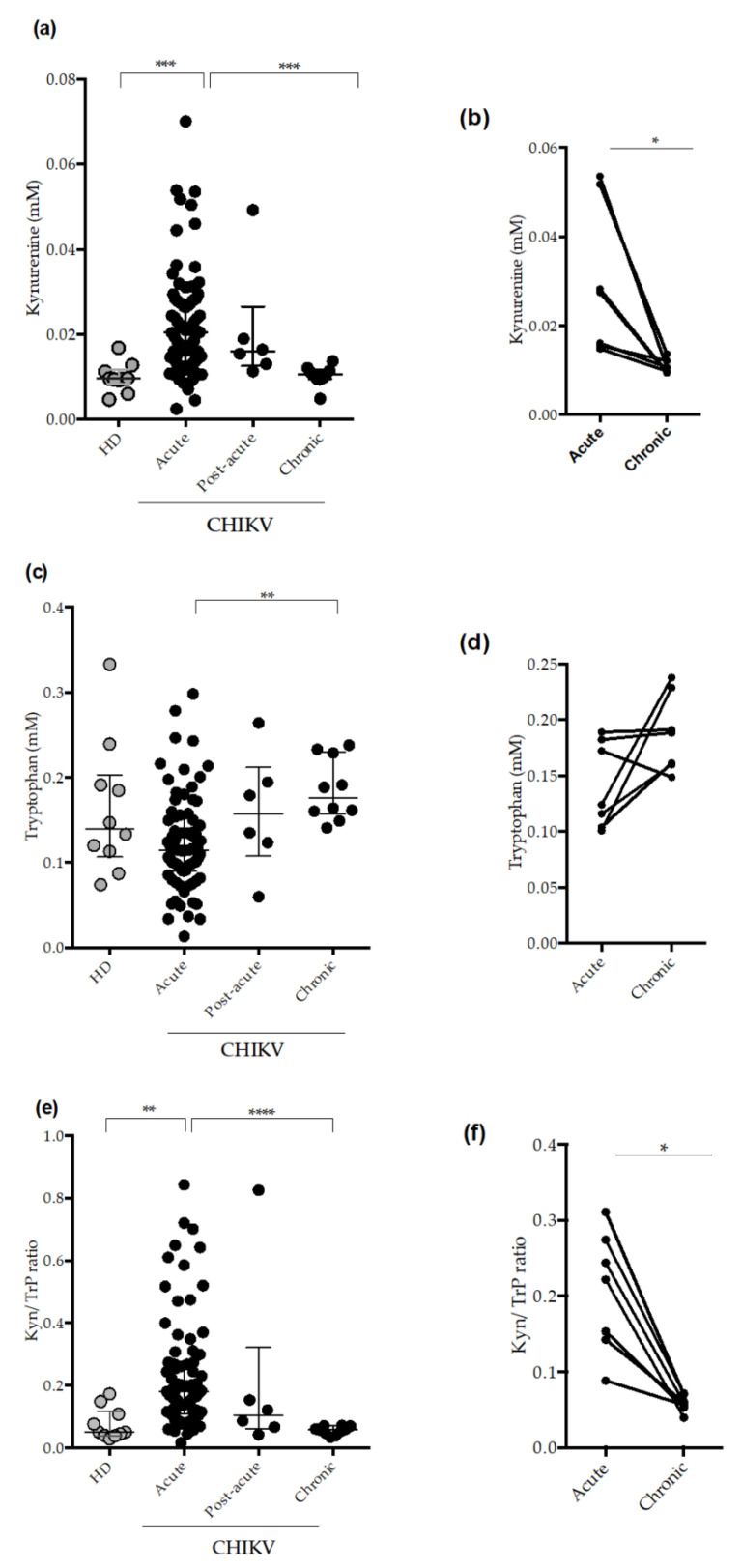
Circulating plasma levels of kynurenine, tryptophan, and Indoleamine 2,3 Dioxygenase 1 (IDO-1) activity in CHIKV-infected patients. (**a**). Quantification of the plasma levels of kynurenine during acute (*n* = 77), post-acute (*n* = 6), and chronic phase (*n* = 10). Healthy donors -HD (*n* = 10). (**b**) Kynurenine levels in paired samples acute vs. chronic. (**c**) Quantification of the plasma levels of tryptophan during acute (*n* = 77), post-acute (*n* = 6), and chronic phase (*n* = 10). Healthy donors-HD (*n* = 10). (**d**) Tryptophan levels in paired samples acute vs. chronic. (**e**) Indoleamine 2,3 Dioxygenase 1 (IDO-1) activity (kyn/Trp ratio) during acute (*n* = 77), post-acute (*n* = 6), and chronic phase (*n* = 10). Healthy donors (HD) (*n* = 10). (**f**) Indoleamine 2,3 Dioxygenase 1 (IDO-1) activity in paired samples acute vs. chronic. For the statistical analysis, the Wilcoxon matched-pairs signed rank, Kruskal–Wallis and Dunn’s multiple comparisons test were performed. * *p* ≤ 0.05, ** *p* ≤ 0.01, *** *p* ≤ 0.001 and **** *p* ≤ 0.0001. Median and interquartile range (IQR).

**Figure 3 pathogens-11-00444-f003:**
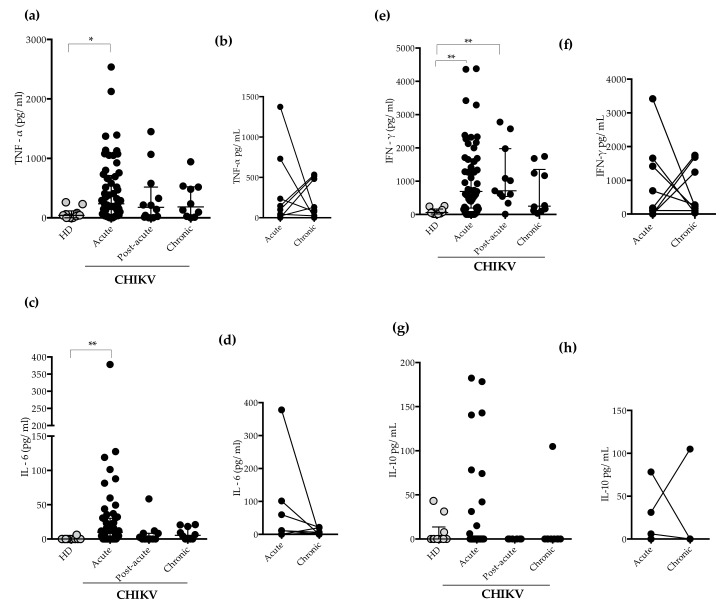
Circulating plasma levels of TNF-α, IL-6 IFN-γ, and IL-10 (pg/mL) in CHIKV-infected patients. (**a**). Quantification of the plasma levels of TNF-α during acute (*n* = 57), post-acute (*n* = 12), and chronic phase (*n* = 10). Healthy donors (HD) (*n* = 10). (**b**) TNF-α levels in paired samples acute vs. chronic. (**c**) Quantification of the plasma levels of IL-6 during acute (*n* = 56), post-acute (*n* = 11), and chronic phase (*n* = 10). Healthy donors (HD) (*n* = 10). (**d**) IL-6 levels in paired samples acute vs. chronic. (**e**) IFN-γ levels during acute (*n* = 53), post-acute (*n* = 11), and chronic phase (*n* = 10). Healthy donors (HD) (*n* = 10). (**f**) IFN-γ levels in paired samples acute vs. chronic. (**g**) Quantification of the plasma levels of IL-10 during acute (*n* = 46), post-acute (*n* = 6), and chronic phase (*n* = 9). Healthy donors (HD) (*n* = 10). (**h**) IL-10 levels in paired samples acute vs. chronic. For the statistical analysis, the Wilcoxon matched-pairs signed rank, Kruskal–Wallis and Dunn’s multiple comparisons test were performed. * *p* ≤ 0.05 and ** *p* ≤ 0.01. Median and interquartile range (IQR).

**Figure 4 pathogens-11-00444-f004:**
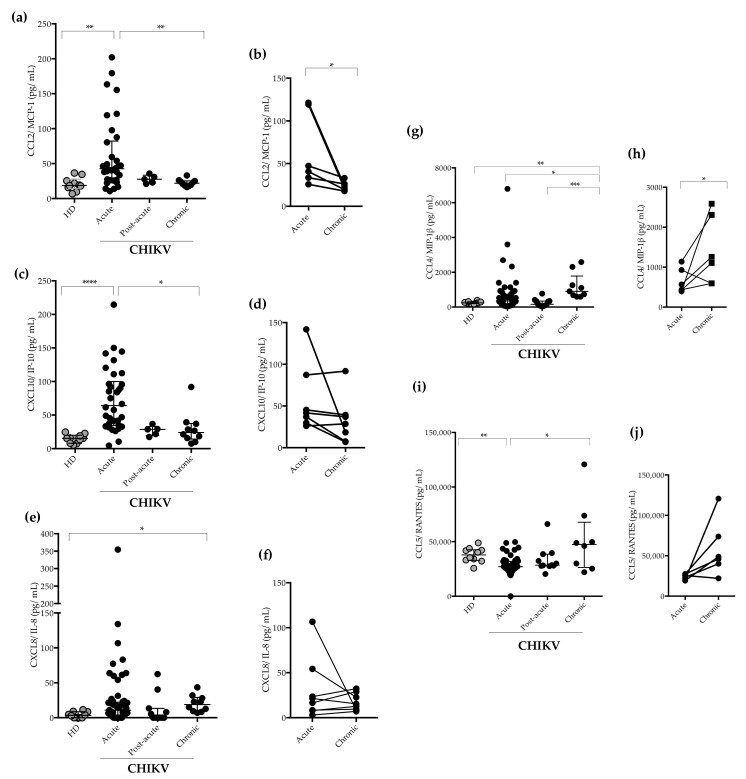
Circulating plasma levels of chemokines CCL2/MCP-1, CXCL10/IP-10, CXCL8/IL-8, CCL4/MIP1-β, and CCL5/RANTES (pg/mL) in CHIKV-infected patients. (**a**) Quantification of the plasma levels of CCL2/MCP-1during acute (*n* = 34), post-acute (*n* = 5), and chronic phase (*n* = 10). Healthy donors (HD) (*n* = 10). (**b**) CCL2/MCP-1 levels in paired samples acute vs. chronic. (**c**) Quantification of the plasma levels of CXCL10/IP-10 during acute (*n* = 34), post-acute (*n* = 5), and chronic phase (*n* = 10). Healthy donors (HD) (*n* = 10). (**d**) CXCL10/IP-10 levels in paired samples acute vs. chronic. (**e**) CXCL8/IL-8 levels during acute (*n* = 60), post-acute (*n* = 11), and chronic phase (*n* = 10). Healthy donors (HD) (*n* = 10). (**f**) CXCL8/IL-8 levels in paired samples acute vs. chronic. (**g**) Quantification of the plasma levels of CCL4/MIP1-β during acute (*n* = 57), post-acute (*n* = 12), and chronic phase (*n* = 9). Healthy donors (HD) (*n* = 10). (**h**) CCL4/MIP1-β levels in paired samples acute vs. chronic. (**i**) Quantification of the plasma levels of CCL5/RANTES during acute (*n* = 49), post-acute (*n* = 11), and chronic phase (*n* = 8). Healthy donors (HD) (*n* = 10). (**j**) CCL5/RANTES levels in paired samples acute vs. chronic. For the statistical analysis, Wilcoxon matched-pairs signed rank, Kruskal–Wallis and Dunn’s multiple comparisons tests were performed. * *p* ≤ 0.05, ** *p* ≤ 0.01, *** *p* ≤ 0.001, and **** *p* ≤ 0.0001. Median and interquartile range (IQR).

**Figure 5 pathogens-11-00444-f005:**
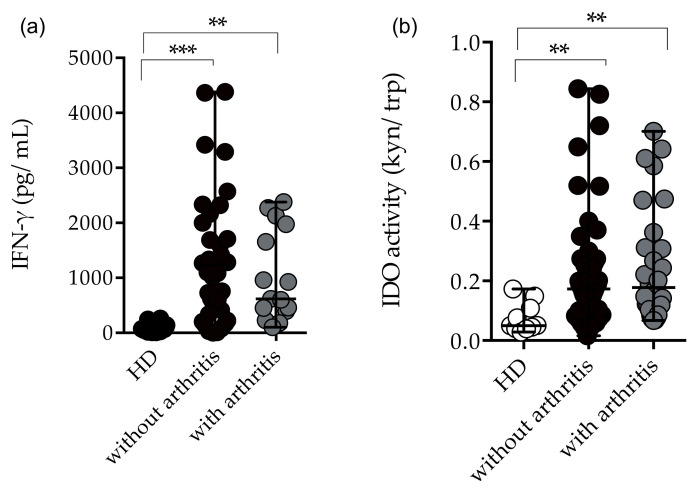
Plasma concentrations of IFN-γ and IDO-1 activity in chikungunya patients without arthritis and with arthritis. (**a**) Plasma concentrations of IFN-γ (pg/mL) in patients without arthritis (*n* = 41) and with arthritis (*n* = 15). Healthy donors (HD) (*n* = 10). (**b**) IDO-1 activity (kyn/Trp ratio) in patients without arthritis (*n* = 54) and with arthritis (*n* = 28). Healthy donors (HD) (*n* = 10). For statistical analysis, the Kruskal–Wallis and Dunn’s multiple comparisons tests were performed. ** *p* ≤ 0.01 and *** *p* ≤ 0.001. Median and interquartile range (IQR).

**Figure 6 pathogens-11-00444-f006:**
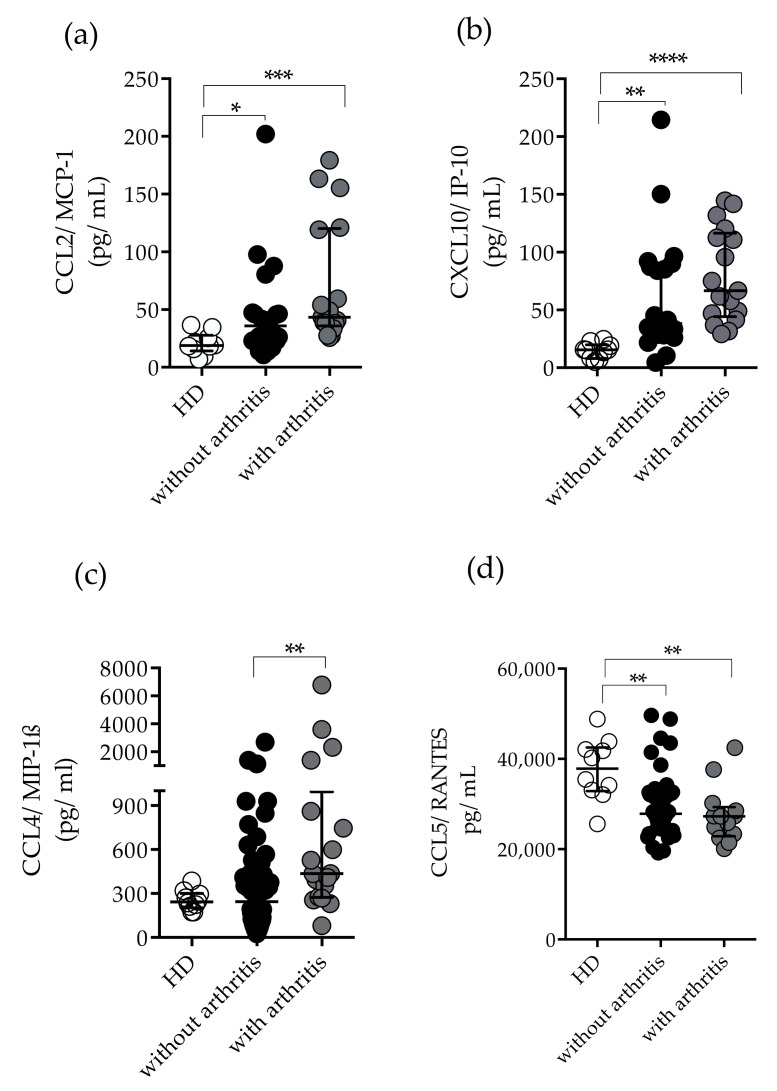
Plasma concentrations of CCL2/MCP-1, CXCL10/IP-10, CCL4/MIP1-β, and CCL5/RANTES in chikungunya patients without arthritis and with arthritis (**a**) Plasma concentrations of CCL2/MCP-1 and (**b**) CXCL10/IP-10 (pg/mL) in patients without arthritis (*n* = 20) and with arthritis (*n* = 17), respectively. Healthy donors (HD) (*n* = 10). (**c**) Plasma concentrations of CCL4/MIP1-β in patients without arthritis (*n* = 50) and with arthritis (*n* = 18), respectively. Healthy donors (HD) (*n* = 10). (**d**) Plasma concentrations of CCL5/RANTES in patients without arthritis (*n* = 43) and with arthritis (*n* = 13), respectively. Healthy donors (HD) (*n* = 10). For the statistical analysis, Kruskal–Wallis and Dunn’s multiple comparison tests were performed. * *p* ≤ 0.05, ** *p* ≤ 0.01, *** *p* ≤ 0.001, and **** *p* ≤ 0.0001. Median and interquartile range (IQR).

**Figure 7 pathogens-11-00444-f007:**
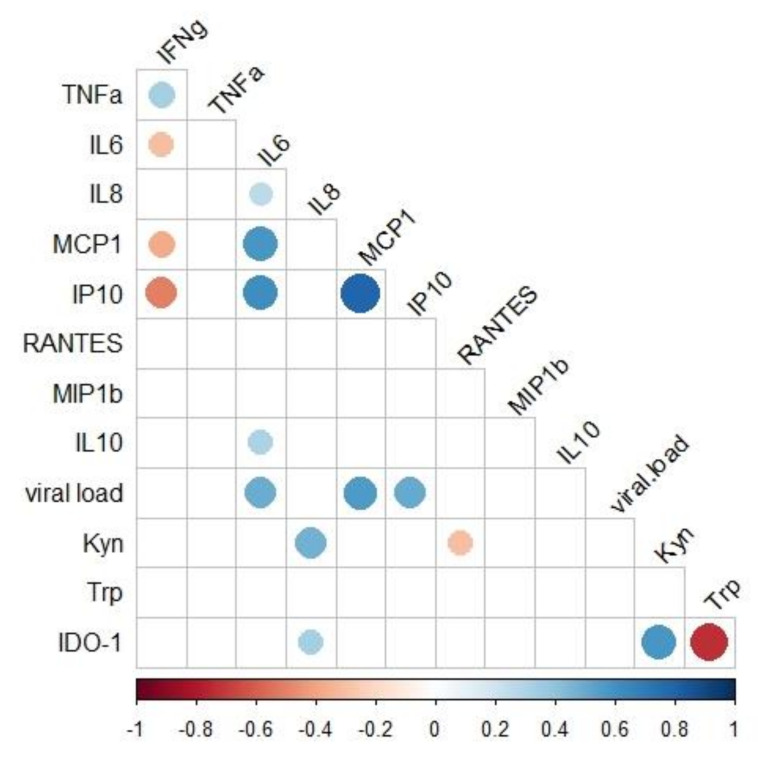
Associations of immunological and virologic factors. Spearman correlation correlogram between cytokines/chemokines, kynurenine, tryptophan, IDO-1 activity, and virologic parameters of confirmed cases of chikungunya. The strength of the correlation between two variables is represented by the color of the circle (only shown if *p* value < 0.05). Colors range from bright blue (strong positive correlation; rs = 1) to bright red (strong negative correlation; rs = −1). Large dots signify higher R values.

**Table 1 pathogens-11-00444-t001:** Demographic and clinical characteristics of chikungunya patients investigated during outbreaks that occurred in RJ, Brazil, 2016–2018.

		Chikungunya *n* (%)	
	Acute Phase *n* = 78	Post-Acute Phase *n* = 12	Chronic Phase *n* = 10
Gender			
Female *n* (%)	45 (57.7)	7 (58.3)	6 (60)
Male	33 (42.3)	5 (41.7)	4 (40)
Age			
Years ^1^	45 (28–57)	54 (36.7–65)	60 (52–66.2)
Days after disease onset	3 (2.0–5.5)	30 (20.5–51)	91 (91–110)
Laboratorial Diagnosis			
RT-PCR	78 (100)	5 (41)	8 (80)
CHIKV Viral Load			
Ct values ^1^	24.8 (18–32.6)	35.1 (26.1–36.6)	30.1 (27.5–30.9)
Viral load _Log_ copies /mL	4.9 (3.1–6.2)	3.0 (2.9–4.1)	4.1 (3.7–4.2)
Anti-CHIKV antibodies			
IgM	35 (44.8)	12 (100)	10 (100)
IgG	6 (7.6)	12 (100)	10 (100)
Signs/Symptoms			
Fever	72 (92)	3 (25)	0
Headache	57 (73)	8 (66)	0
Running nose	6 (7.6)	0	0
Cough	7 (8.9)	0	0
Retroorbital pain	38 (48)	4 (33)	0
Myalgia	54 (69)	6 (50)	5 (50)
Low back pain	48 (61)	4 (33)	0
Arthralgia	73 (93)	12 (100)	10 (100)
Polyarthritis	25 (32)	4 (33)	6 (60)
Edema	40 (51)	7 (58)	6 (60)
Anorexia	53 (67)	5 (41)	0
Prostration	59 (75)	6 (50)	5 (50)
Dizziness	23 (29)	2 (16)	0
Nausea	40 (51)	3 (25)	0
Vomiting	17 (21)	2 (16)	0
Abdominal pain	13 (16)	0	0
Pruritus	34 (43)	2 (16)	0
Exanthema	41 (52)	2 (16)	0
Paresthesia	8 (10)	2 (16)	2 (20)
Conjunctival hyperemia	26 (33)	0	0
Hospitalization			
	3 (3.5)	0	0

^1^ Values are expressed in median and interquartile range (IQR).

## Data Availability

The detailed data can be provided from the authors upon request.

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
