# Peer review of "Increased Indoleamine 2,3-Dioxygenase 1 (IDO-1) Activity and Inflammatory Responses during Chikungunya Virus Infection"

_pathogens, 2022, doi:10.3390/pathogens11040444_

Round 1
Reviewer 1 Report
The authors analyzed changes in IDO-1 activity, cytokine and chemokine patterns in CHIKV infected patients. Authors observed increased levels in IDO-1 activity, TNF- α, IL-6, IFN- γ, and CCL2/MCP-1 and CXCL10/IP-10 at the acute phase while CCL4/MIP1β and CCL5/RANTES were increased in chronic phase of infection. Overall, the study is well performed, scientifically acceptable and clinically relevant. The study design and methods are described well and cohort is sufficient.
Some minor concerns should be addressed
- In the discussion the implications for the study was not discussed such as the effect of the upregulated cytokine/chemokines. Discussion could be concise.
- Study period month not mentioned from 2016 – 2018
- L. 486-488 repeated information from introduction.
- Figure 7 what is the purpose use of DPI?
- Can you elaborate on what are the effect of IDO-1 activity during acute and chronic phase of infection? or during arthritis. It is not clear
- I found some spelling errors in the text.
Author Response
"Please see the attachment."

Reviewer 2 Report
The MS is potentially relevant to the field of chikungunya and immunology. This is a very important study to show the immune system markers in CHIKV patients. Obviously, this is a preliminary study due to the necessity to understand the real mechanisms of action to immunology and CHIKV disease. However, this MS adds value and scientific relevance in this field for the scientific community. The MS is very much organized and clear.
Author Response
"Please see the attachment.

Reviewer 3 Report
Souza at al describes the activity of IDO-1, an important extrahepatic Trp-degrading enzyme largely known to be a biomarker of immune activation, during the CHIKV infection through measurements of [Kyr]/[Trp] ratio. In addition, the authors show the plasma levels of several pro- and anti-inflammatory cytokine/chemokines from plasma of CHIKV-infected patients collected at different periods of infection. Overall, the manuscript sounds novel and interesting and contributes to understand the immune imbalance triggered by CHIKV as well as the contribution of some pro-inflammatory cytokines to the establishment of the chronic disease. However, there are some major issues that authors should address:
- In Fig.2 , the authors describes an extremely high concentration of plasma Kynurenin and Tryptophan levels, around milimolar range, in both the healthy controls and CHIKV-patients. These values are not in agreement with the reported serum levels for these two molecules in the literature, which are in micromolar range (see the references indicated in the text). Moreover, as the authors are quantifying a molecule concentration in body fluid, it would ideally be presented as ug/mL or equivalent instead of molarity.
- The authors declared there are some limitation in their work, mainly regarding the small number of paired samples, which is clearly understandable and accepted. However, alongside the manuscript, the authors show the Kyr, Trp and cytokines levels of the plasma panel stratified according to the days post-infection and also the correlation of these biomarkers in paired samples, which were stratified as acute/post-acute and chronic. Comparing the individual data and the paired samples, one can see that some biomarkers present different behavior between acute-phase and post-acute phase, therefore, the comparison made in paired samples might demonstrate an artifact or at least do not exhibit the individual variations observed in individual analysis. I suggest the authors re-stratify these samples in acute-chronic and post-acute-chronic in order to be more realistic with the data observed in individual samples, even if some statistical analysis could not be made.
- The discussion section is extremely large and should be more concise and focus on the most interesting data of the work that is the IDO-1 role in the metabolic and immune response activities.
- The english should be improved and several misspelling should be corrected.
Author Response
"Please see the attachment

Round 2
Reviewer 3 Report
The manuscript was considerably improved.